# Antiperiodontitis Effects of *Siegesbeckia glabrescens* In Vitro

**DOI:** 10.3390/antiox12020471

**Published:** 2023-02-13

**Authors:** Arce Defeo Bellere, Duna Yu, Sarang Oh, Myeongju Kim, Jeyong Jung, Minzhe Fang, Shengdao Zheng, Tae-Hoo Yi

**Affiliations:** 1Graduate School of Biotechnology, Kyung Hee University, Yongin-si 17104, Republic of Korea; 2Snowwhitefactory Co., Ltd., 807 Nonhyeon-ro, Gangnam-gu, Seoul 06032, Republic of Korea

**Keywords:** *Siegesbeckia glabrescens*, periodontitis, *Porphyromonas gingivalis*, antioxidants, biofilms

## Abstract

*Siegesbeckia glabrescens* is generally grown in fields or roadsides in Korea and used for the treatment of inflammatory diseases. The effects of *S. glabrescens* on periodontitis are unknown. In this study, we determined the effects of an *S. glabrescens* 30% EtOH extract (SGE) on periodontitis and analyzed the antioxidant activity (DPPH, ABTS, and SOD), antimicrobial (disc diffusion, MIC, and MBC), inhibition of GTFs, biofilm formation, and the anti-inflammation of lipopolysaccharide from *P. gingivalis* (LPS-PG)-induced primary equine periodontal ligament fibroblasts (PDLFs). We report that SGE increased DPPH, ABTS, and SOD antioxidant activities in a dose-dependent manner. SGE caused a clear zone with a diameter of 15 mm or more against periodontal pathogens. SGE (2.50 mg/mL) inhibited GTFs and biofilm by 89.07% and 85.40%, respectively. SGE treatment (100 µg/mL) also significantly decreased the secretion of inflammatory mediators in sensitized PDLF, including cytokines and matrix metalloproteinase (MMP)-3, -8, -9, and -13. Overall, we confirmed that SGE had excellent antioxidant, antimicrobial, and anti-inflammatory effects against periodontal pathogens. These results suggest that it has the potential to develop as a prophylactic agent for periodontitis.

## 1. Introduction

Oral health enhances nutrient intake and digestion, and it is vital to maintain systemic health. Major diseases representing oral health conditions include dental caries, periodontal disease, and oral cancer [1]. If left untreated, these diseases not only result in tooth loss; they also cause serious effects throughout the body and, in some cases, lead to death.

Periodontitis, a type of gum disease, is a chronic inflammatory disease with symptoms, such as red, bleeding, and swollen gums, caused by an organized bacterial community (biofilm) called a plaque [2]. Higher risk factors of periodontal disease are associated with diabetes, medication, age, hereditary, poor oral hygiene, smoking, and stress [3].

The excessive proliferation of pathogenic bacteria present in dental plaque and the gingival sulcus is considered a direct cause of periodontal disease. Among these bacteria, the ones that are recognized as the cause of periodontal disease are mainly Gram-negative anaerobic bacteria, such as *Porphyromonas gingivalis*, *Aggregatibacter actinomycetemcomitans*, *Treponema denticola*, *Prevotella intermedia*, and *Fusobacterium nucleatum*. Socransky et al. [4]. classified subgingival biofilm bacteria into five complexes, namely red, orange, yellow, green, and purple, based on the results of staining reaction, colony morphology, and pigment production [5]. Among the five complexes, the red complex colonizes during the late stage of periodontitis, has the strongest pathogenicity, and is highly correlated with clinical indicators of periodontal diseases, such as deep periodontal pockets and bleeding [6].

*Streptococcus* spp. is a resident bacterium in the oral cavity and is one of the causes of periodontitis. Endogenous enzymes in cells are activated by surface proteins of these bacteria, which decompose tissue components and pathological changes [7]. One of the representative intermediates that plays a role in periodontitis is matrix metalloproteinase (MMP). MMP is an enzyme that is involved in various inflammatory diseases in the body and decomposes extracellular substrates, such as gelatin and collagen, and it is related to intraoral diseases, such as periodontal disease, tooth decay, oral cancer, cleft palate, and axillary cyst [8].

*Siegesbeckia glabrescens* is a dicotyledonous plant and an annual herb belonging to the Asteraceae family. It grows wild throughout Korea, Japan, Taiwan, and China and is a plant that generally grows in fields or roadsides in Korea [9]. The whole plant, except for the root, is primarily used as a medicine, and it is known to be effective for diseases such as quadriplegia, arthritis, stroke, headache, high blood pressure, dizziness, jaundice, acute hepatitis, swelling, and skin itching [10]. According to a recent study, *S. glabrescens* exhibits excellent biological activities, including anti-arthritis, anti-inflammatory, antiphotoaging, antibacterial, and antimelanogenesis effects [11,12,13,14]. Among the active compounds in *S. glabrescens* 30% EtOH extract (SGE), kirenol is known to have excellent biological activity; however, there are no reports regarding the antiperiodontitis properties of *S. glabrescens*. 

Although *S. glabrescens* is known for its significant benefits, the biological potential for periodontitis-related disorders is unexplored. Therefore, in the present study, we contributed to the development of natural periodontitis inhibitors by evaluating the antimicrobial, anti-inflammation, and antioxidant effects of SGE.

## 2. Materials and Methods

### 2.1. Preparation of S. glabrescens Extraction

Dried *S. glabrescens* (180 g, cultivated in Chungju-si, Chungcheongbuk-do, Republic of Korea) was mixed with 1.8 L of 30% EtOH. The *S. glabrescens* 30% EtOH extract (SGE) was prepared by extraction under pressure at 80 °C for 2 h. The resulting extract was filtered through a 0.45 μm filter paper (Whatman, USA); concentrated to 15 Brix, using a rotary evaporator (EYELA WORLD—Tokyo Rikakikai Co., LTD., Tokyo, Japan); and then spray-dried.

### 2.2. HPLC Analysis of SGE

SGE was prepared at a concentration of 2 mg/mL in 50% methanol. Serial dilutions (2.5, 25, 125, 250, 500, and 1000 µg/mL) of the standard compound (kirenol) in methanol were prepared. HPLC was performed on an Ultimate 3000 LC system, using P580 and UVD100 detectors (Thermo Fisher Scientific Inc., Waltham, MA, USA). Chromatographic separation was performed on an Inno C-18 column (5 μm, 4.6 × 250 nm; Young Jin). The column temperature was 40 °C, the flow rate was 1.0 mL/min, the analysis time was 40 min, and the injection volume was 10 µL. To verify the validity of the method, the experiment was conducted based on the “Guideline for Validation of Test Methods such as Drugs” (Ministry of Food and Drug Safety).

### 2.3. Disc Diffusion Assay

Dental caries and periodontal pathogens were purchased from the Korean Agricultural Culture Collection (KACC) and Korean Collection for Type Cultures. The disc diffusion method was performed based on the method of Mounyr Balouiri et al. [15]. Dental caries (100 μL) and periodontal pathogens (1 × 10^6^ CFU/mL) were aliquoted onto an agar plate and smeared with a sterile cotton swab. Then 100 μL of each sample was dissolved in sterile distilled water, loaded onto 8 mm paper discs (Advantec Toyo Kaisha, Ltd., Tokyo, Japan), and dried for 15 min. After drying, the paper discs were placed on an agar plate, using sterile tweezers, and incubated for 24 h at 37 °C. Anaerobic bacteria were incubated in an MGC AnaeroPack Rectangular Jar. Antimicrobial activity was assessed by measuring the size of the inhibition zone.

### 2.4. Minimum Inhibitory Concentration and Minimum Bactericidal Concentration Assay 

Minimum inhibitory concentration (MIC) and minimum bactericidal concentration (MBC) were performed using the 96-well microplate dilution method described in the Clinical and Laboratory Standards Institute guidelines. After incubation (24 h, 37 °C), the absorbance at 600 nm was measured using a microplate reader (Molecular Devices Filter Max F5; San Francisco, CA, USA). Using an inoculation loop, a portion of each well was streaked on an agar plate suitable for dental caries and periodontal pathogens and then incubated for 24 h at 37 °C. The colonies were evaluated to determine the MIC and MBC.

### 2.5. Scanning Electron Microscope Analysis

*S. mutans* untreated and treated with the MBC of candidate samples was centrifuged at 4000 rpm for 15 min to collect pellets, and then 1 mL of Karnovsky’s fixative was added for protein correction and fixed at 4 °C for 24 h. The fixed bacteria were washed three times every 5 min with 0.05 M sodium cacodylate buffer. After washing, the bacteria were fixed with 1% osmium tetroxide at 4 °C for 1 h, and the bacteria were washed three times every 5 min with sterile distilled water. The washed bacteria were dehydrated every 10 min with an increasing concentration of ethanol at each time (30, 50, 70, 80, 90, and 100% ethanol; 100% ethanol was repeated three times). After dehydration, the bacteria were added to 100 μL of hexamethyldisilazane (HMDS) and dried for 24 h. Then carbon tape was attached to a stub on which the dried bacteria were mounted. Finally, using an Ion Sputter Coater, plasma was created on the surface of the nonconductive sample to create a metal film. The bacteria were imaged using an SU8010 scanning electron microscope (Hitachi, Japan).

### 2.6. Inhibition of Glucosyltransferase Activity

To measure the inhibition of glucosyltransferases (GTFs) activity, the Mukasa [16] method was used. *S. mutans* was cultured for 24 h in a 37 °C incubator and centrifuged at 4000 rpm for 15 min. The pH of the supernatant was adjusted to 7.0, using 0.1 M NaOH. After calibration, 0.02% sodium azide 1 mL was added to prepare a GTF-coenzyme solution. Then 1 mL of GTF-coenzyme solution, the diluted sample (0.08–2.5 mg/mL), and 1 mL of 2.0% sucrose were added and placed in an incubator at 37 °C for 24 h. The absorbance was measured at 600 nm, using a UV–Vis Spectrophotometer (Mecasys CO, Ltd., Daejon, Korea). The inhibition rate of GTFs’ activity (GIR (%)) was determined by setting the initial absorbance before the reaction as I and the absorbance at 24 h after the reaction as T. When the measured GIR value was 50% or less, the extract was considered to have excellent GTF-inhibitory activity.
GIR (%) = (T/I) × 100

### 2.7. Inhibition of Biofilm Formation

A biofilm-formation assay was performed based on the method of Sara Moataz Zayed et al. [17]. An inoculum was prepared by diluting *S. mutans* with 0.85% NaCl solution so that it could be inoculated into each well at 1 × 10^6^ CFU/mL. *S. mutans*, combined with the SGE (0.08–2.5 mg/mL), was incubated for 24 h at 37 °C. The supernatant was removed, each well was washed three times with 200 μL of sterile distilled water, and the microplate was dried with a blow-dryer. Then 50 μL of 0.1% crystal violet solution was added to the biofilm created at the bottom of the microplate and stained for 15 min. The supernatant was removed, and the biofilm was washed three times with 200 μL of sterile distilled water and then dried. Finally, 200 μL of 95% EtOH was added to dissolve the dyed biofilm, and the absorbance was measured at 595 nm, using a microplate reader.

### 2.8. Inhibition of Acid Production

Organic acids produced by *S. mutans* have a direct effect on teeth. Therefore, it was confirmed that SGE could inhibit acid formation. *S. mutans* was diluted with 0.85% NaCl to prepare an inoculum (1 × 10^6^ CFU/mL). Then 0.5 mL of the inoculum was added to 15 mL conical tubes containing the diluted extract. After culturing for 24 h in an incubator at 37 °C, the pH was measured using a pH meter (Seven Compact Duo S213, Switzerland), and the growth of the bacteria was determined at 600 nm, using a UV–Vis Spectrophotometer.

### 2.9. Total Phenolic and Flavonoid Contents

The total phenolic content of SGE was determined using the Folin–Ciocâlteu colorimetric method [18]. The absorbance was measured for 50% methanol blanks at 625 nm, using a microplate reader. Total phenolic content is expressed as mg gallic acid equivalents per gram of dry extract. The total flavonoid content of SGE was measured by the aluminum chloride colorimetric method [19]. The absorbance was measured at 450 nm, using a microplate reader. Total flavonoid content is expressed as mg quercetin equivalents per gram of dry extract.

### 2.10. 1,1-Diphenyl-2-Picrylhydrazyl and 2,2′-Azino-Bis-(3-Ethylbenzothiazoline)-6-Sulfonic Acid Radical Scavenging Assay

The antioxidative activity of SGE was determined using 1,1-diphenyl-2-picrylhydrazyl (DPPH). A 0.2 mM DPPH solution was prepared using 100% methanol, 40 µL of sample, and 160 µL of DPPH solution that was added to each well of a 96-well microplate, which was then incubated at 37 °C for 30 min (under dark conditions). After the reaction, the absorbance was measured at 595 nm, using a microplate reader. The antioxidative activity of SGE was determined using 2,2′-azino-bis-(3-ethylbenzothiazoline)-6-sulfonic acid (ABTS). A 2.5 mM ABTS solution, 1 mM 2,2′-azobis(2-amidinopropane) dihydrochloride (AAPH), and 150 mM sodium chloride were mixed and incubated overnight. Then 4 μL of sample and 196 μL of ABTS solution were added to each well of a 96-well microplate and incubated at 37 °C for 30 min (under dark conditions). After the reaction, the absorbance was measured at 405 nm, using a microplate reader. The radical scavenging activity of the SGE was determined using the following formula:DPPH and ABTS radical inhibition (%) = [(OD_0_ − OD_x_)/OD_0_] × 100
where OD_0_ is the negative control, and OD_x_ is the SGE and ascorbic acid tested at various concentrations (1–100 µg/mL).

### 2.11. Superoxide Dismutase-Like Assay

The superoxide dismutase (SOD)-like activity was measured by the method of Marklund and Marklund [20]. The SOD-like activity of SGE was measured for pyrogallol (Sigma-Aldrich, Co., St. Louis, MO, USA). Briefly, 10 μL of SGE, 130 μL of Tris-HCl buffer, and 10 μL of 7.2 mM pyrogallol (prepared immediately before use) were added to each well of a 96-well microplate and incubated at 20 °C for 10 min. Then 10 μL of 1 N HCl was added to terminate the reaction, and the absorbance was measured at 420 nm, using a microplate reader. The radical scavenging activity of the SGE was determined using the following formula:SOD-like activity (%) = [(OD_0_ − OD_x_)/OD_0_] × 100
where OD_0_ is the negative control, and OD_x_ is the SGE and ascorbic acid tested at various concentrations (1–100 µg/mL).

### 2.12. Cytotoxicity

Primary equine periodontal ligament fibroblast (PDLF) cells were provided by the Korean Cell Bank (Seoul, Korea). The cells were cultured in a humidified incubator at 37 °C, containing 5% CO_2_, using DMEM medium (Hyclone Laboratories Inc., Logan, UT, USA) supplemented with 10% heat-inactivated FBS and 1% antibiotics and antifungal agents (Hyclone Laboratories Inc.). PDLF cells were lipopolysaccharide from *P. gingivalis* (LPS-PG)-induced to cause an inflammatory response. PDLF cells were sensitized with 1 µg/mL LPS-PG at 80% confluence for 24 h. An aliquot of 10 µM dexamethasone (positive control) or SGE (1–100 µg/mL) was diluted in serum-free medium and incubated for the same duration as the LPS-PG treatment. After 24 h of treatment with LPS-PG, 1 µg/mL MTT was added to the culture and incubated for 4 h. The medium was discarded, followed by the addition of DMSO to solubilize formazan. The optical density was recorded at a wavelength of 595 nm.

### 2.13. NO Assay

NO production was measured in LPS-PG-induced PDLF cells. After 24 h of SGE and LPS-PG exposure, the secretion of NO was quantified in the cell culture supernatant. Then 100 µL of cell culture supernatant was incubated with 100 µL of Griess reagent, which is a mixture of 1% sulfanilamide in 5% phosphoric acid and 0.1% naphthyl ethylenediamine dihydrochloride (1:1 ratio). The plate was incubated for 10 min at 37 °C, and the absorbance was measured at 595 nm.

### 2.14. Enzyme-Linked Immunosorbent Assay 

SGE and LPS-PG were incubated with PDLF cells for 24 h, and the supernatant was collected. The concentration of MMP-3, MMP-8, MMP-9, and MMP-13 protein in the supernatant was measured using an ELISA kit (R&D Systems Inc.) according to the manufacturer’s instructions.

### 2.15. Statistical Analysis

The data were analyzed using a Statistical Analysis System (GraphPad Prism 5). All the quantitative data were expressed as means ± standard deviation. The significance of differences was determined using a one-way analysis of variance (ANOVA) with Duncan’s test for multiple comparisons. Statistical significance was considered at *p* < 0.05. All experiments were performed in triplicate.

## 3. Results

### 3.1. Quantitative Analysis of Kirenol from SGE

A quantitative analysis for kirenol in the SGE revealed a peak at the same retention time of the standard kirenol (kirenol, 24.610 min; SGE, 24.627 min) (Figure 1). The kirenol content of the SGE was 10.69 ± 0.07 mg/mL.

### 3.2. Antimicrobial Activity

The antimicrobial activity of SGE was tested against dental caries and periodontal pathogens by using the disc diffusion method. The SGE showed a clear zone with a diameter of 15 mm or more against dental caries (Table 1) and periodontal pathogens (Table 2).

### 3.3. MIC and MBC Assays

MIC and MBC assays were used to determine the concentration of SGE required against dental caries and periodontal pathogens. The results indicated that the MBC of the SGE required to kill dental caries pathogens was 0.31 mg/mL or more (Table 3). In addition, the MBC of the SGE required to kill periodontal pathogens was 0.40 mg/mL or higher (Table 4).

### 3.4. Morphological Changes in Streptococcus Mutans

The cell wall is a rigid structure on the outside of the cell membrane and plays an important role in protecting the cell from the outside and maintaining its shape. After treatment of *Streptococcus mutans* KACC 16833 with SGE, morphological changes in the bacterial cell walls were observed using a scanning electron microscope. Untreated *S. mutans* KACC 16833 had smooth surfaces (Figure 2a), whereas *S. mutans* KACC 16833 treated with SGE exhibited severe cell wall destruction (Figure 2b).

### 3.5. Inhibition of GTFs’ Activity

Biofilm is the cause of periodontal inflammation. The inhibition of GTFs involved in biofilm formation was evaluated, and the results are shown in Figure 3. SGE exhibited 32.18, 51.48, 72.49, 83.57, and 89.07% inhibitory activity at concentrations of 0.16, 0.31, 0.63, 1.25, and 2.50 mg/mL, respectively.

### 3.6. Inhibition of Biofilm Formation

We determined whether SGE was effective at inhibiting biofilm formation, and the results are shown in Figure 4. SGE exhibited 58.61, 69.35, 85.04, 85.11, 85.20, and 85.40% inhibitory activity at concentrations of 0.08, 0.16, 0.31, 0.63, 1.25, and 2.50 mg/mL, respectively.

### 3.7. Inhibition of Acid Production

To determine whether SGE inhibits acid production by *S. mutans,* measurements were made using a pH meter and a UV–Vis Spectrophotometer. As shown in Figure 5, when *S. mutans* was cultured in Trypticase Soy Yeast (TSY) broth, the pH after culture was 4.035. Acid production was inhibited by SGE at concentrations above the MIC (0.63, 1.25, and 2.50 mg/mL, including concentrations of 0.31 mg/mL) and acid production was observed at concentrations below the MIC (0.16 and 0.08 mg/mL). The growth of *S. mutans* was also inhibited at the concentration in which acid production was inhibited and acid production increased proportionally with the growth of *S. mutans*.

### 3.8. Total Phenolic and Flavonoid Contents

SGE exhibited a total phenolic content of 59.76 ± 1.76 mg GAE/g extract and a total flavonoid content of 0.25 ± 0.06 mg QE/g extract.

### 3.9. Antioxidant Activity

To determine the antioxidant properties of SGE, its radical scavenging and oxidation activities were measured using a DPPH, ABTS, and SOD assay. Ascorbic acid was used as a positive control, and we found that SGE inhibited DPPH (Figure 6a) and ABTS (Figure 6b) radicals in a dose-dependent manner. The DPPH and ABTS IC_50_ values of SGE were 1.99 μg/mL and 0.39 μg/mL, respectively, whereas they were 6.10 μg/mL and 0.17 μg/mL for ascorbic acid, respectively. SOD-like activity increased concomitantly with increased SGE (Figure 6c). In particular, SGE exhibited high activity of 80% or more at 100 μg/mL.

### 3.10. Cytotoxicity and Regulation of NO Production of SGE

The toxicity of SGE on PDLF cells was determined using an MTT assay. As shown in Figure 7a, the LPS-PG treatment of PDLF cells slightly decreased the survival of cells by 34.6% compared with the normal group; however, supplementing the assay with 10 µM dexamethasone and SGE restored cell viability (Figure 7a).

When NO is produced excessively, it can enhance the inflammatory response and promote cancer. As shown in Figure 7b, compared with the untreated PDLF cells, NO synthesis was increased by 1177.1% in the LPS-PG-treated cells after 24 h. However, in the SGE-treated groups, NO secretion decreased by 7.4%, 14.8%, and 27.8% at 1, 10, and 1000 µg/mL, respectively, when compared to the LPS-PG-treated controls.

### 3.11. Effect of SGE on the Secretion of MMP-3, MMP-8, MMP-9, and MMP-13

To determine the inhibitory effect of SGE on collagen degradation, we quantified the secreted levels of MMP-3, -8, -9, and -13 in LPS-PG-treated cell culture supernatants by using ELISA kits. Treatment with LPS-PG increased MMP-3, -8, -9, and -13 protein levels by 887.2%, 556.9%, 548.5%, and 3381.3%, respectively (Figure 8). In contrast, treatment with SGE resulted in a dose-dependent inhibitory effect, in which the secretion of MMP-3, -8, -9, and -13 was inhibited by 67.1%, 42.3%, 56.1%, and 78.9%, respectively.

## 4. Discussion

Herein, the antioxidant effect of SGE and its antibacterial effects against dental caries and periodontal pathogens were examined. The inhibitory effects of SGE on inflammatory expression in PDLF cells and biofilm formation by *S. mutans* were confirmed, indicating its preventative activity against periodontitis.

Periodontal diseases, specifically periodontitis, are caused by pathogenic bacterial species located in the subgingival niche [21]. Hatipoglu et al. [22] reported that several risk factors contribute to the progression of periodontal disease, of which the main cause is microbial dental plaque. Therefore, inhibiting the growth of periodontal pathogens will effectively prevent or improve periodontal status. To determine the antimicrobial mechanisms inactivated by SGE, we tested the extract against dental caries and periodontal pathogens, using the disc diffusion method. As shown in Table 1 and Table 2, SGE inhibited dental caries and periodontal pathogens with a diameter of 15 mm or more. An examination of the morphological changes using SEM revealed that *S. mutans* was torn and destroyed (Figure 2).

*Porphyromonas gingivalis* is a Gram-negative oral anaerobe that is involved in the pathogenesis of periodontitis, which is an inflammatory disease that destroys the tissues supporting the tooth and eventually leads to tooth loss [23]. It was hypothesized that *P. gingivalis* and NO levels increase in patients with periodontitis, depending on the severity of the inflammation. Therefore, to compare the effect of SGE on the growth of microorganisms, we performed MIC and MBC assays. The MIC and MBC values for SGE against *P. gingivalis* were at least 0.20 and 0.40 mg/mL, respectively (Table 3 and Table 4). Matejka et al. found increased levels of amino acids related to NO in inflammatory periodontal disease. Similarly, we demonstrated that SGE effectively decreased the secretion of NO (Figure 7b). These findings support the potential of SGE as a prophylactic agent for periodontal inflammation.

Periodontitis is caused by bacteria present in the form of biofilms, rather than planktonic bacteria. Recent studies using molecular biological methods have reported that oral diseases are the result of interactions between various bacteria in biofilms [24]. Therefore, inhibiting the formation of biofilm is effective for the treatment and prevention of oral diseases. In the present study, we showed that SGE inhibited biofilm formation in a dose-dependent manner. In particular, at 2.5 mg/mL SGE, an inhibition rate of 85.4% was observed (Figure 4). Glucosyltransferases (GTFs) are key enzymes that contribute to the development of oral biofilms. The GTFs of *S. mutans* are recognized as one of the causes of dental caries and periodontal inflammation because they contribute to the formation of dental plaque and the establishment of *S. mutans* on the tooth surface. Herein, we evaluated the effect of SGE on inhibiting GTFs’ activity, which is involved in biofilm formation. SGE at a concentration of 2.5 mg/mL inhibited GTFs’ activity by 89.07% (Figure 3). In addition, acid production was inhibited at concentrations above the MIC (Figure 5). As a result, SGE exhibits excellent activity as a biofilm inhibitor.

Active oxygen generated during the cause or process of periodontal disease can damage tissues to cause periodontal disease or to aggravate ongoing periodontal disease [25]. Oxidative stress resulting from free oxygen causes periodontal disease by the irreversible destruction of lipids, proteins, sugars, and DNA, which are cellular components [26]. Therefore, antioxidants play a very important role in periodontitis. In the present study, we found that SGE has antioxidant properties by scavenging DPPH and ABTS radicals and exhibiting SOD-like activity. In particular, SGE showed inhibitory effects on DPPH and ABTS radicals with IC_50_ values of 1.99 μg/mL and 0.39 μg/mL, respectively (Figure 6a,b). Jacoby and Davis [27] reported that antioxidants, such as vitamin E, ascorbic acid, and selenium, are effective at treating superoxide resulting from inflammation associated with periodontal disease. Interestingly, SGE exhibited SOD-like inhibitory activity in a dose-dependent manner (Figure 6c).

An imbalance in the regulation of MMP activity results in tissue destruction, fibrosis, and degradation of the extracellular matrix, which occurs at various stages of disease progression [28]. MMP-3 and MMP-8, besides being involved in periodontal pathology, are also associated with cardiovascular pathology and diabetes [29]. MMP-9 expression is associated with damage to periodontal tissue during the active stages of periodontitis [30]. MMP-13 expression is important to the proliferation of periodontal damage, the progression of attachment loss, and the deepening of periodontal pockets [31]. Therefore, we focused on these targets for SGE’s action by measuring the secretion of MMP-3, -8, -9, and -13 in PDLF cells after LPS-PG induction. The increase in MMP-3, -8, -9, and -13 levels was significantly reduced in the presence of SGE and was dose-dependent (Figure 8). This suggests that SGE ameliorates periodontitis by inhibiting MMP-3, -8, -9, and -13 expressions in PDLF cells.

## 5. Conclusions

In conclusion, SGE has the potential to inhibit the GTFs’ activity and biofilm formation induced by *S. mutans* by targeting specific pathways. SGE inhibited NO and MMPs in PDLF cells treated with *P. gingivalis* and regulated the production of reactive oxygen species as a strong antioxidant both directly and indirectly. Therefore, SGE is potentially valuable as a new prophylactic agent for periodontitis, with antioxidant, antibacterial, and anti-inflammatory effects. It is clear that, for SGE, further experimental work making use of, for example, in vivo systems, clinical, etc., will be necessary. Therefore, in the future, we will have to set up various comprehensive databases linked with periodontitis biomarkers.

## Figures and Tables

**Figure 1 antioxidants-12-00471-f001:**
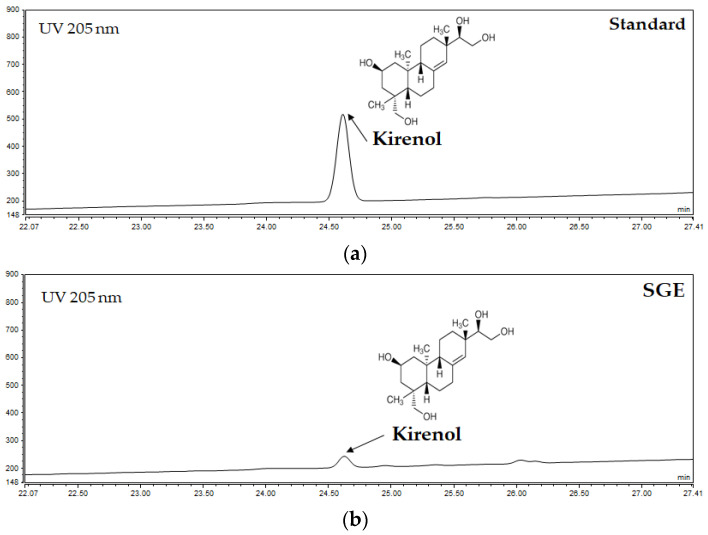
RP-HPLC analysis of kirenol standard (**a**) and SGE (**b**).

**Figure 2 antioxidants-12-00471-f002:**
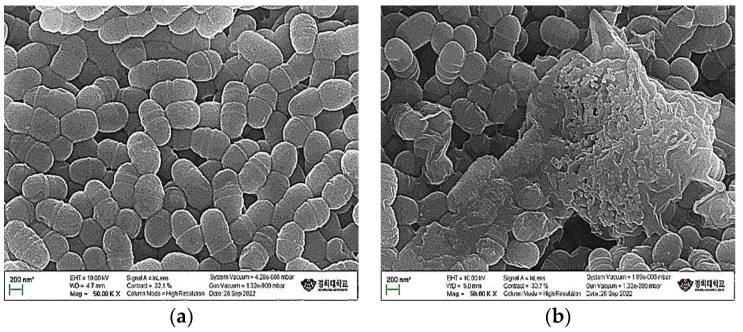
Scanning electron micrographs of untreated *S. mutans* and after 24 h of treatment with SGE at a fixed magnification. *S. mutans* KACC 16833 (**a**). Treated with SGE (**b**).

**Figure 3 antioxidants-12-00471-f003:**
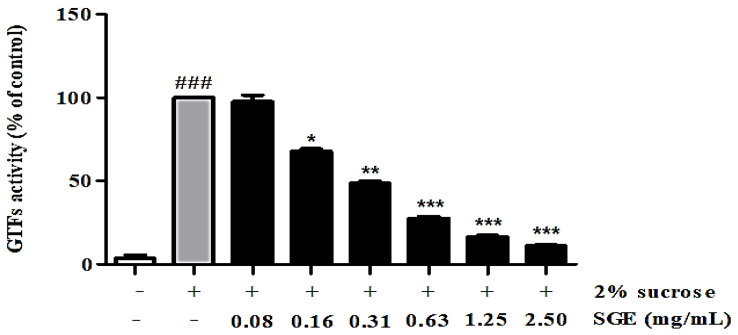
Inhibition of GTFs’ activity of SGE. GTFs’ inhibitory activity was measured using a spectrophotometer of water-insoluble glucan produced by *S. mutans*, using sucrose as a substrate. Glucosyltransferase was pretreated with the indicated concentrations of each sample for 24 h. Values are mean ± SD; ### *p* < 0.001 vs. non-treated group, * *p* < 0.05, ** *p* < 0.01, and *** *p* < 0.001 vs. 2% sucrose-treated group.

**Figure 4 antioxidants-12-00471-f004:**
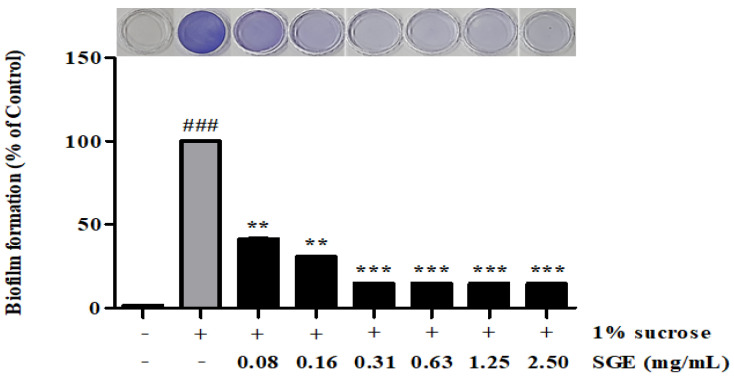
Inhibition of biofilm formation of SGE. Using 1% sucrose as a substrate, the biofilm-formation inhibitory effect of SGE on the biofilm produced by *S. mutans* was confirmed. After treatment at the indicated concentration for each extract, incubated for 24 h, the biofilm formation was measured using a microplate reader. Values are mean ± SD. ### *p* < 0.001 vs. non-treated group, ** *p* < 0.01, *** *p* < 0.001 vs. 1% sucrose-treated group.

**Figure 5 antioxidants-12-00471-f005:**
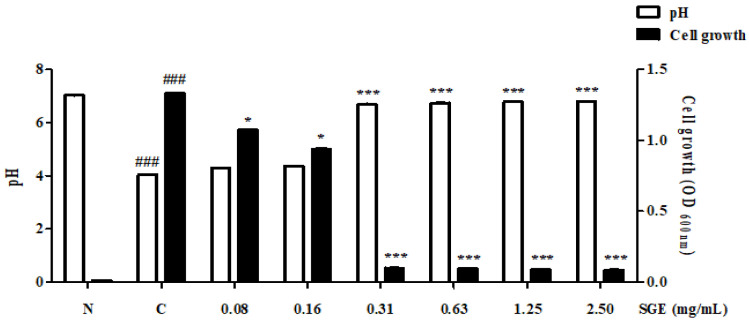
Inhibition of acid production of SGE. After treating *S. mutans* with the indicated concentration of SGE for 24 h, the amount of acid production was measured using a pH meter, and bacterial growth was measured using a spectrophotometer. Values are mean ± SD; ### *p* < 0.001 vs. non-treated group, * *p* < 0.05, and *** *p* < 0.001 vs. 1% sucrose-treated group.

**Figure 6 antioxidants-12-00471-f006:**
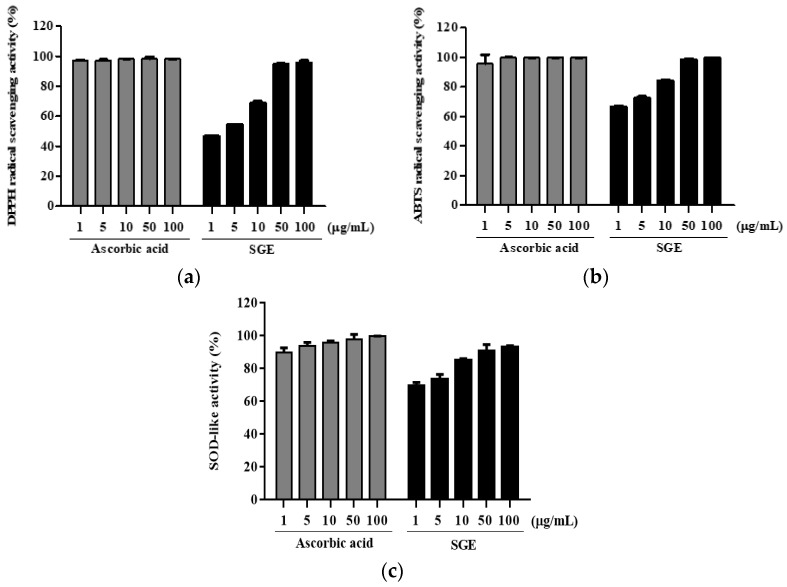
DPPH, ABTS, and SOD-like antioxidation activity of SGE. DPPH radical scavenging (**a**), ABTS^+^ cation scavenging (**b**), and SOD-like activity of SGE (**c**). Ascorbic acid was used as a positive control. Data are expressed as mean ± SD of the results of three replicates.

**Figure 7 antioxidants-12-00471-f007:**
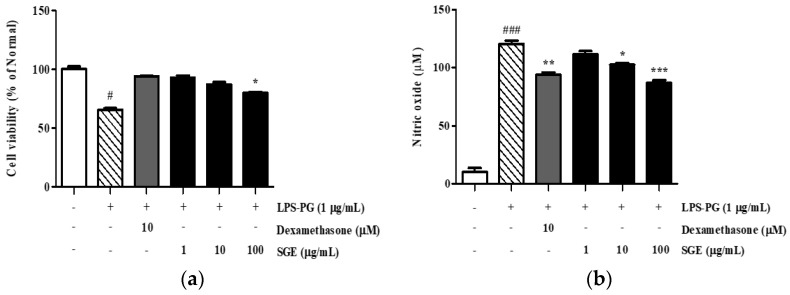
Cell viability and NO production in LPS-PG-induced PDLF cells treated with SGE. Effect SGE on cell viability (**a**). Effect of SGE on NO production (**b**). Dexamethasone was used as a positive control. Values are mean ± SD; # *p* < 0.05, ### *p* < 0.001 vs. non-treated group, * *p* < 0.05, ** *p* < 0.01, and ****p* < 0.001 vs. LPS-PG-treated group.

**Figure 8 antioxidants-12-00471-f008:**
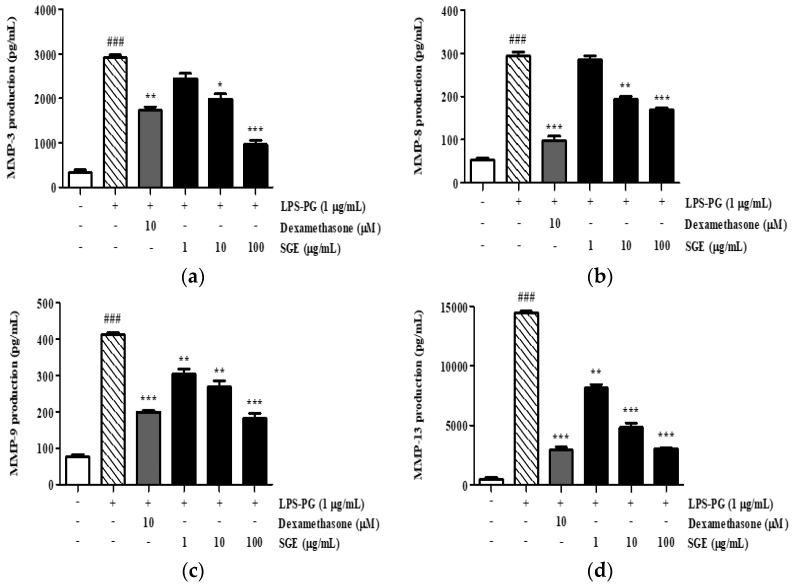
SGE inhibited the secretion of MMP-3, -8, -9, and -13 in LPS-PG-induced PDLF cells. MMP-3 (**a**), MMP-8 (**b**), MMP-9 (**c**), and MMP-13 (**d**) secretion by LPS-PG-induced PDLF cells. Dexamethasone was used as a positive control. Values are mean ± SD; ### *p* < 0.001 vs. non-treated group, * *p* < 0.05, ** *p* < 0.01, and *** *p* < 0.001 vs. LPS-PG-treated group.

**Table 1 antioxidants-12-00471-t001:** Antimicrobial activity of SGE against oral *Streptococci*.

Indicators	Inhibition Zone	
	Amoxicillin	SGE
*Streptococcus mutans* KACC 16833	++ **	++
*Streptococcus sanguinis* KACC 11301	++	++
*Streptococcus downei* KACC 13827	++	++
*Streptococcus gordonii* KACC 13829	++	++
*Streptococcus ferus* KACC 13881	++	++
*Streptococcus mitis* KACC 16832	++	++

++ **: over 15 mm.

**Table 2 antioxidants-12-00471-t002:** Antimicrobial activity of SGE against periodontal pathogens.

Indicators	Inhibition Zone	
	Amoxicillin	SGE
*Porphyromonas gingivalis* KCTC 5352	++ **	++
*Treponema denticola* KCTC 15104	++	++
*Campylobacter gracilis* KCTC 15224	++	++
*Campylobacter rectus* KCTC 5636	++	++
*Fusobacterium nucleatum* KCTC 2640	++	++
*Parvimonas micra* KCTC 15021	++	++
*Prevotella intermedia* KCTC 15693	++	++
*Prevotella nigrescens* KCTC 15081	++	++
*Aggregatibacter actinomycetemcomitans* KCTC 2581	++	++
*Eikenella corrodens* KCTC 15198	++	++

++ **: over 15 mm.

**Table 3 antioxidants-12-00471-t003:** MIC and MBC analysis of SGE against oral *Streptococci*.

Indicators	SGE (mg/mL)	
	MIC	MBC
*S. mutans* KACC 16833	≥0.63	1.30
*S. sanguinis* KACC 11301	≥0.40	0.80
*S. downei* KACC 13827	≥0.20	0.40
*S. gordonii* KACC 13829	≥0.20	0.31
*S. ferus* KACC 13881	≥1.30	3.00
*S. mitis* KACC 16832	≥0.40	0.80

**Table 4 antioxidants-12-00471-t004:** MIC and MBC analysis of SGE against periodontal pathogens.

Indicators	SGE (mg/mL)	
	MIC	MBC
*P. gingivalis* KCTC 5352	≥0.20	0.40
*T. denticola* KCTC 15104	≥0.20	0.40
*C. gracilis* KCTC 15224	≥0.31	0.63
*C. rectus* KCTC 5636	≥0.31	0.63
*F. nucleatum* KCTC 2640	≥0.20	0.40
*P. micra* KCTC 15021	≥0.31	0.63
*P. intermedia* KCTC 15693	≥0.31	0.63
*P. nigrescens* KCTC 15081	≥0.31	0.63
*A. actinomycetemcomitans* KCTC 2581	≥0.20	0.40
*E. corrodens* KCTC 15198	≥0.20	0.40

## Data Availability

The data presented in this study are available in this paper.

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
