# Peer review of "Antiperiodontitis Effects of Siegesbeckia glabrescens In Vitro"

_antioxidants, 2023, doi:10.3390/antiox12020471_

Round 1
Reviewer 1 Report
The manuscript entitled “Antiperiodontitis effects of Siegesbeckia glabrescens”, authored Bellere et al., deals with evaluation of biological properties of Siegesbeckia glabrescens, including usefulness in the treatment of periodontitis. The topic of this manuscript is important and current, and results could be interesting for readers. However, some changes have to be entered into the revised version of the manuscript before it can be further processed:
1. Kirenol is an active ingredient contained in the raw material. Its solubility in water is limited, slightly soluble in ethanol. I'm afraid that using 30% ethanol may not be enough to get the best extract. 70% ethanol is much more preferred.
2. In the abstract, there is no information about the performance of microbiological activity tests
3. Chapter 3.9 - more valuable than showing that ascorbic acid inhibits the radical 100% is showing the IC50 value. Figure 6 is redundant since some values have been indicated in the text.
4. There is no discussion about the possibility of application in practice. What pharmaceutical form should be proposed?
Author Response
The manuscript entitled “Antiperiodontitis effects of Siegesbeckia glabrescens”, authored Bellere et al., deals with evaluation of biological properties of Siegesbeckia glabrescens, including usefulness in the treatment of periodontitis. The topic of this manuscript is important and current, and results could be interesting for readers. However, some changes have to be entered into the revised version of the manuscript before it can be further processed:
- Kirenol is an active ingredient contained in the raw material. Its solubility in water is limited, slightly soluble in ethanol. I'm afraid that using 30% ethanol may not be enough to get the best extract. 70% ethanol is much more preferred.
Thank you for your valuable comment where it is desirable to use higher content of ethanol to extract Kirenol. However, since our research aims to develop products such as healthy functional food, we considered the cost and efficacy aspects of our product. Based on our findings, and utilization of 30% ethanol extract we were able to divulge that 30% ethanol extract had anti-inflammatory and anti-microbial effects and detected kirenol which is presumed to be an effective substance. Additionally, from an industrial point of view, lowering ethanol content to 30% will lead to higher chances of product development. Thus, we believed that 30% ethanol extraction is ideal and feasible for our experiment.
- In the abstract, there is no information about the performance of microbiological activity tests
Thank you for your suggestion, and agree with you. However, in the antibacterial experiment, we determined that GTFs and biofilm were closely related to periodontitis. Please understand that the rest of the antibacterial test results were reflected in the methods and results of the manuscript.
- Chapter 3.9 - more valuable than showing that ascorbic acid inhibits the radical 100% is showing the IC50 value. Figure 6 is redundant since some values have been indicated in the text.
We deeply understand your comment. We utilized ascorbic acid as a positive control in DPPH, ABTS, and SOD wherein we wrote each IC50 result in the manuscript. The inhibition rate of ascorbic acid was too high that the graphical value looks alike, but it was utilized in an independent experiment (ABTS, DPPH, and SOD). Thus, we would greatly appreciate it if you could consider looking at these findings.
- There is no discussion about the possibility of application in practice. What pharmaceutical form should be proposed?
We deeply agree with your suggestion. In this study, the potential of S. glabrescens for periodontal drug development was elucidated as antibiotic and anti-inflammatory drug for periodontal-associated diseases es or oral hygiene products. Furthermore, it is expected to be utilized as an oral drug or functional product and be applicable after confirmation of its toxicity and in vivo tests in the future.
We deeply appreciate for all your help in revising this manuscript. Thanks to the reviewer, we were able to make much more upgraded manuscript. It was such an honor to receive a review from you.
Reviewer 2 Report
Dear Authors,
Thank you very much for submitting you manuscript to the prestigious journal Antioxidants which has an impressive IF.
I hope that my pieces of advice and remarks will be useful in order to increase its’ scientific quality.
1. Line 3 – please write authors names in the same manner
2. Line 5 – please check carefully the way in which the affiliation are written
3. Line 39 – please rephrase
4. Line 53 – Please rephrase in a different way. It’ s not an absolute cause.
5. A table including abbreviations as they apear in the text might be useful
6. Line 391 – I would suggest to rephrase the Conclusions section in such a way that it will reflect better the clinical impact of your findings
Best regards!
Author Response
Dear Authors,
Thank you very much for submitting you manuscript to the prestigious journal Antioxidants which has an impressive IF.
I hope that my pieces of advice and remarks will be useful in order to increase its’ scientific quality.
- Line 3 – please write authors names in the same manner
We appreciate your comment. All authors names in lines 3 of the revised manuscript were corrected in the same manner.
- Line 5 – please check carefully the way in which the affiliation are written
Thank you for your comment. We again meticulously checked the affiliation of the authors of the manuscript.
- Line 39 – please rephrase
Thank you for your advice. Following the reviewer's advice, the contents of line 39 of the revised manuscript were rewritten.
- Line 53 – Please rephrase in a different way. It’ s not an absolute cause.
Thank you for your recommendation. The expression was changed to ‘Streptococcus spp., which is a resident bacterium in the oral cavity and is one of the causes of periodontitis’ in the revised manuscript.
- A table including abbreviations as they apear in the text might be useful
Thank you for pointing it out. To reflect your opinion, we have written the scientific name of the first abbreviation in the table.
- Line 391 – I would suggest to rephrase the Conclusions section in such a way that it will reflect better the clinical impact of your findings
Thank you for your guidance. The conclusions section has been phrased in a way that reflects its clinical impact.
Best regards!
We deeply appreciate for all your help in revising this manuscript. Thanks to the reviewer, we were able to make much more upgraded manuscript. It was such an honor to receive a review from you.
Reviewer 3 Report
The manuscript entitled "Antiperiodontitis effects of Siegesbeckia glabrescens" brings new inside in periodontitis treatment which is a problematic disease regarding the efficiency of existing treatments. Therefore every new efficient treatment couldbe a real benefit for periodontitis and periodontal disease patients.
Abstract
Please organize your abstract in backgrounds, Methods, Results and Conclusions.
Introduction
The aim of the study should be more detailed and should describe how the authors want to contribute to the development of periodontitis inhibitors using SGE.
The paragraph between line 348 and line 349 (figure 8) has to be moved in Results chapter.
In the discussion chapter you only need to refer to the figures/tables you already described in Results chapter.
Please indicate at the end of sentence the references (line 327).
Author Response
The manuscript entitled "Antiperiodontitis effects of Siegesbeckia glabrescens" brings new inside in periodontitis treatment which is a problematic disease regarding the efficiency of existing treatments. Therefore every new efficient treatment couldbe a real benefit for periodontitis and periodontal disease patients.
Abstract
- Please organize your abstract in backgrounds, Methods, Results and Conclusions.
We appreciate your suggestion. The purpose of the study, experimental methods, results, and conclusions have already been written in the abstract as concisely as possible. Please review it again.
Introduction
- The aim of the study should be more detailed and should describe how the authors want to contribute to the development of periodontitis inhibitors using SGE.
Thank you for your valuable suggestion. We have added revisions on line 70 to facilitate the o how we would like to contribute to the development of periodontitis inhibitors using SGE
- The paragraph between line 348 and line 349 (figure 8) has to be moved in Results chapter.
Thanks for your point. Figure 8 was intended to be inserted in the result section, but due to problems such as the size of the figure, the figure was moved to the discussion section. Thank you for your understanding.
- In the discussion chapter you only need to refer to the figures/tables you already described in Results chapter.
Thank you for your advice. We framed the discussion to illustrate that SGE is effective in preventing periodontitis by referring once more to the figures and tables of the results. It's a reiteration, but it's something I think is important. Please look at it with a wide tolerance.
- Please indicate at the end of sentence the references (line 327).
We appreciate your recommendation. This is our research result. There cannot be references.
We deeply appreciate for all your help in revising this manuscript. Thanks to the reviewer, we were able to make much more upgraded manuscript. It was such an honor to receive a review from you.
Reviewer 4 Report
The subject is very interesting, the paper is well written the present study, we contributed to the development of natural periodontitis inhibitors by evaluating the effects of SGE. SGE has the potential to inhibit GTFs activity and biofilm formation induced by S. mutans by targeting specific pathways. SGE inhibited NO and MMPs in P. gingivalis-treated PDLF cells, and regulated the production of reactive oxygen species as a strong antioxidant both directly and indirectly. Therefore, SGE is potentially valuable as a new prophylactic agent for periodontitis
Establishes the originality of the research aims by demonstrating the need for investigations in the topic area
· Title
It should state in the title that it is an experimental in vitro study.
Antiperiodontitis is an unaccepted term and my opinion is that it should be modified.
· Incomplete introduction. The authors seem to have little knowledge of the subject
Siegesbeckia glabrescens is a dicotyledonous plant and an annual herb belonging to the Asteraceae family. It grows wild throughout Korea, Japan, Taiwan, and China, and is a plant that generally grows in fields or roadsides in Korea It should give more content about its biochemical characteristics and profile.
· Material and methods are well developed the methods are used appropriate
· The results are supported by graphs that improve the interpretation of the results.
It should highlight the strong points and add the limitations of the study
Author Response
The subject is very interesting, the paper is well written the present study, we contributed to the development of natural periodontitis inhibitors by evaluating the effects of SGE. SGE has the potential to inhibit GTFs activity and biofilm formation induced by S. mutans by targeting specific pathways. SGE inhibited NO and MMPs in P. gingivalis-treated PDLF cells, and regulated the production of reactive oxygen species as a strong antioxidant both directly and indirectly. Therefore, SGE is potentially valuable as a new prophylactic agent for periodontitis
Establishes the originality of the research aims by demonstrating the need for investigations in the topic area
Title
1. It should state in the title that it is an experimental in vitro study.
Thanks for the point. But I don't think it's necessary to state that it's an in vitro experiment. I think the title now covers a wider range of meanings. Please understand with a broad mind.
2. Antiperiodontitis is an unaccepted term and my opinion is that it should be modified.
We appreciate your comment. In this regard, we believe that antiperiodontitis can be used since we work on the causative agent of periodontitis, and it has been used in other studies. For instance, the study of Jee et.al. (2009), entitled: “Antiperiodontitis Effects of Magnolia bindii Extract on Ligature-Induced Periodontitis in Rats” also utilized the word periodontitis. We hope you kindly look into consideration on this.
- Incomplete introduction. The authors seem to have little knowledge of the subject
Siegesbeckia glabrescens is a dicotyledonous plant and an annual herb belonging to the Asteraceae family. It grows wild throughout Korea, Japan, Taiwan, and China, and is a plant that generally grows in fields or roadsides in Korea It should give more content about its biochemical characteristics and profile.
Thank you for your comment. A description of the general taxonomy and origin of Siegesbeckia glabrescens. If you tell me what kind of professional knowledge would be good to enter, I am willing to reflect on it.
4. Material and methods are well developed the methods are used appropriate
The results are supported by graphs that improve the interpretation of the results.
Thank you for a good look.
5. It should highlight the strong points and add the limitations of the study
We appreciate your comment. We added some content to the Conclusion. See lines 402-404.
We deeply appreciate for all your help in revising this manuscript. Thanks to the reviewer, we were able to make much more upgraded manuscript. It was such an honor to receive a review from you.
Reviewer 5 Report
This study assessed the effects of Siegesbeckia glabrescens 30% ethanol alcohol (EtOH) extract on periodontitis. Siegesbeckia glabrescens (SG) has been used in traditional oriental medicine for the treatment of allergic and inflammatory diseases. The extracts and bioactive components of SG have anti-inflammatory, anti-allergic, and anticancer activities. The manuscript contains five keywords, eight figures, four tables, and thirty-one references. Overall, it is a correct, complete, and well-conducted paper, although some slight remarks are made on different sections of the manuscript.
Keywords
The manuscript presents five keywords. For keywords, where possible, please use Medical Subject Headings terms (MeSH Terms). Initially, only “Periodontitis” and “Porphyromonas gingivalis” are MeSH terms. Some alternative MeSH terms proposed could be “antioxidants” better than “antioxidant” or “biofilms” rather than “biofilm formation”. Nevertheless, these suggestions about keywords are optional, not mandatory.
Other manuscript sections
There are some unexplained abbreviations. Abbreviations and acronyms, even well-known, should be explained the first time they are used.
Line 277. As the abbreviation "TSY" is only used once, please consider substituting it with the full name of the culture medium.
References
Total number of the manuscript references: 31.
The reference format almost matches the journal’s reference format, according to the ACS style guide. Two different formats are used for the journal name (abbreviated journal name and full journal name). The journal's guidelines recommend the use of abbreviated journal names. Please, use the same format for all the references.
For further information about the reference format proposed by the journal, please, consult the following link: https://www.mdpi.com/journal/antioxidants/instructions
Figures
Total number of the manuscript figures: 8.
The figures have appropriate figure legends. However, please consider explaining the abbreviations in the figure legends.
Tables
Total number of the manuscript tables: 4.
The tables have appropriate titles and information. Nevertheless, please, in each table, a footer explaining the abbreviations could be included.
Author Response
This study assessed the effects of Siegesbeckia glabrescens 30% ethanol alcohol (EtOH) extract on periodontitis. Siegesbeckia glabrescens (SG) has been used in traditional oriental medicine for the treatment of allergic and inflammatory diseases. The extracts and bioactive components of SG have anti-inflammatory, anti-allergic, and anticancer activities. The manuscript contains five keywords, eight figures, four tables, and thirty-one references. Overall, it is a correct, complete, and well-conducted paper, although some slight remarks are made on different sections of the manuscript.
Keywords
The manuscript presents five keywords. For keywords, where possible, please use Medical Subject Headings terms (MeSH Terms). Initially, only “Periodontitis” and “Porphyromonas gingivalis” are MeSH terms. Some alternative MeSH terms proposed could be “antioxidants” better than “antioxidant” or “biofilms” rather than “biofilm formation”. Nevertheless, these suggestions about keywords are optional, not mandatory.
Thank you for your suggestion. We totally agree with your opinion. We modified some keywords with MeSH terms suggested by reviewers.
Other manuscript sections
There are some unexplained abbreviations. Abbreviations and acronyms, even well-known, should be explained the first time they are used.
Line 277. As the abbreviation "TSY" is only used once, please consider substituting it with the full name of the culture medium.
Thank you for your observation. We modified the full name of the culture medium.
References
Total number of the manuscript references: 31.
The reference format almost matches the journal’s reference format, according to the ACS style guide. Two different formats are used for the journal name (abbreviated journal name and full journal name). The journal's guidelines recommend the use of abbreviated journal names. Please, use the same format for all the references.
For further information about the reference format proposed by the journal, please, consult the following link: https://www.mdpi.com/journal/antioxidants/instructions
Thank you for your comment. The changes were reflected in the manuscript.
Figures
Total number of the manuscript figures: 8.
The figures have appropriate figure legends. However, please consider explaining the abbreviations in the figure legends.
Thank you for pointing it out. Most of the abbreviations are explained in "Introduction and Materials and Methods". However, if you continue to want to include the abbreviation, we will whenever reflect.
Tables
Total number of the manuscript tables: 4.
The tables have appropriate titles and information. Nevertheless, please, in each table, a footer explaining the abbreviations could be included.
Thank you for your suggestion. To reflect your opinion, we have written the scientific name of the first abbreviation in the table. However, if you continue to want to include the abbreviation, we will whenever
We deeply appreciate for all your help in revising this manuscript. Thanks to the reviewer, we were able to make much more upgraded manuscript. It was such an honor to receive a review from you.
Round 2
Reviewer 1 Report
I appreciate the contribution in preparing the answer, however, none of the changes I suggested have been implemented.
1. In the abstract, information on microbiological tests should be introduced.
2. Figure 6 is redundant. Antioxidant activity results should be reported as IC50.
Author Response
- Thanks for the point. Corrected in the abstract. Please see line 17.
- Thanks for your comments. However, we also reflected the figure and IC50 when publishing papers in this existing journal. Please note.
Reviewer 4 Report
Adding the word in vitro gives a global view of the paper to the readers.
Although this term has been used, it is not considered as such in periodontal disease guidelines and I think it is excessive in the title.
Author Response
We appreciate your comment.
Changed the title from "Antiperiodontitis effects of Siegesbeckia glabrescens " to "Antiperiodontitis effects of Siegesbeckia glabrescens in vitro".
Thanks to. sincerely
Round 3
Reviewer 1 Report
Accept in present form